# Addressing Uncertainty in Tool Wear Prediction with Dropout-Based Neural Network

Arup Dey [1,*], Nita Yodo [2,*], Om P. Yadav [3], Ragavanantham Shanmugam [4] and Monsuru Ramoni [1]

1   School of Engineering, Math, and Technology, Navajo Technical University, Crownpoint, NM 87313, USA
2   Department of Industrial and Manufacturing Engineering, North Dakota State University, Fargo, ND 58102, USA
3   Department of Industrial & Systems Engineering, North Carolina Agricultural and Technical State University, Greensboro, NC 27411, USA; oyadav@ncat.edu
4   Department of Engineering Technology, Fairmont State University, Fairmont, WV 26554, USA
*   Correspondence: a.dey@navajotech.edu (A.D.); nita.yodo@ndsu.edu (N.Y.)

**Abstract:** Data-driven algorithms have been widely applied in predicting tool wear because of the high prediction performance of the algorithms, availability of data sets, and advancements in computing capabilities in recent years. Although most algorithms are supposed to generate outcomes with high precision and accuracy, this is not always true in practice. Uncertainty exists in distinct phases of applying data-driven algorithms due to noises and randomness in data, the presence of redundant and irrelevant features, and model assumptions. Uncertainty due to noise and missing data is known as data uncertainty. On the other hand, model assumptions and imperfection are reasons for model uncertainty. In this paper, both types of uncertainty are considered in the tool wear prediction. Empirical mode decomposition is applied to reduce uncertainty from raw data. Additionally, the Monte Carlo dropout technique is used in training a neural network algorithm to incorporate model uncertainty. The unique feature of the proposed method is that it estimates tool wear as an interval, and the interval range represents the degree of uncertainty. Different performance measurement matrices are used to compare the proposed method. It is shown that the proposed approach can predict tool wear with higher accuracy.

**Keywords:** Monte Carlo dropout; uncertainty; tool wear; principal component analysis; interval prediction

## 1. Introduction

An intelligent manufacturing system is a technology-driven approach that utilizes sensors, the Internet of Things (IoT), artificial intelligence, and data analysis algorithms to improve the efficiency and effectiveness of production processes [1]. In modern manufacturing sectors, such as aerospace and automobile, diverse types of sensors (e.g., vibration and acoustic sensors) are installed to collect real-time data for health monitoring and maintenance decision-making. Different data analytic approaches are being applied to use the collected sensor data for supporting the decision-making processes in engineering maintenance and prognostic applications by predicting different target variables from the data. Tool failure is one of the common reasons for the quality degradation of machined parts and the increase in the downtime of manufacturing systems. Tool flank wear is a common tool failure that affects tool and workpiece properties [2]. The data analytic algorithms are applied in tool wear estimation to avoid unscheduled failures, predict tool changing time, and produce high-quality parts.

The commonly used methods for tool wear prediction can be classified into three broad categories: physics-based, data-driven, and hybrid methods [3]. The domain knowledge and understanding, failure mechanism, and a series of requisite assumptions are required to formulate a physics-based model [4]. Taylor's tool life equation and Usui's wear rate model are two examples of the physics-based model [3]. The accuracy of a physics-based

method can be improved by increasing the understanding of a system. The physics-based method required fewer experimental data. Nevertheless, developing a physics-based model for a complex system is not always possible, as in-depth knowledge about the system is required for this and complex interconnection among different components of a system. On the other hand, the use of data-driven methods do not require in-depth knowledge of the physics of a system [4]. Data-driven methods can capture the degradation or failure behavior using various statistical and learning techniques from condition monitoring data of the essential parameters. Support vector machine (SVM), decision tree, random forest, and artificial neural networks (NNs) are well-known data-driven methods used extensively. A large volume of relevant historical data is required to train a data-driven model. Data-driven models can be modified easily to capture non-linearity and apply to different systems. Often, the prediction accuracy depends on the quantity and quality of the data used. Hybrid methods are developed by integrating physics-based and data-driven approaches to provide more realistic and accurate predictions. The efforts to combine these two methods aim to overcome the weaknesses of using any single method [4]. However, with the significant advancement in data collection caused by sensor technology and IoT, data-driven approaches are becoming more popular for monitoring and predicting tool conditions [5].

The purpose of using data-driven models is to predict a target variable for an unknown situation by collecting data utilizing several types of sensors. The performance of a model is usually evaluated by comparing the predicted value of a target with the true (actual) value, whenever the true value is available. In most cases, the existing methods provide predictions as deterministic point values, assuming that these models predict with high accuracy. However, deterministic predictions are not always appropriate in real-life situations such as engineering and business applications. The deterministic prediction may lead to infeasibility or poor performance [6]. The uncertainty arises due to incomplete information and the random nature of a system [7]. The sources of uncertainty have a significant impact on processes, products, and collected data. Therefore, a target variable's interval prediction or probability distribution estimation is preferable over point value estimation [8]. Additionally, several assumptions considered to develop a data-driven model are potential sources of uncertainty. For example, the assumption of linearity and limited historical data to represent a system by a data-driven model are a few sources of uncertainty. Consequently, the tool wear prediction ignoring these sources of uncertainty directly impacts product quality, experiment results, production cost, and financial decisions. The cost of ignorance of these sources of uncertainty is high, leading to wrong decisions. It is, therefore, crucial to incorporate uncertainty in developing prediction models to leverage robust decisions. The uncertainty can be reduced by collecting more high-quality data as they are reducible. However, this is not always a practical solution since collecting additional data may increase experiment costs and time.

The accuracy and reliability of these data-driven prediction methods are influenced by uncertainty. This uncertainty is caused by the randomness of data, the noise in data collection, and model assumptions [9]. Therefore, the model uncertainty and randomness (and/or noise) in data are considered the two most important sources of uncertainty in data-driven models [10]. Model uncertainty arises due to errors in model selection, lack of sufficient training data, model bias, and model assumptions [11]. Most of these existing data-driven prediction methods provide point estimates, which is not realistic in the decision-making process [8].

A few data-driven methods are developed that consider the uncertainty in making predictions. Hwang et al. [12] proposed a model for interval prediction by a neural network (NN). However, their proposed model is not always practical to predict the interval because it is required to calculate the inverse of a matrix. This method is often computationally expensive, and the inverse exists only for a positive definite matrix. Gaussian process regression (GPR) is a machine learning technique to propagate input uncertainty into a predicted output. The predicted output is represented by a probability distribution [13].

The prior knowledge and specifications about the shape of a model can be incorporated into GPR as a kernel function. Heskes [14] proposed a bootstrapping sampling method for estimating prediction intervals from an ensemble of NNs. The bootstrapping method is very useful for small data sets, but it is assumed that prediction by the ensemble of neural networks is an unbiased estimator of true value. In addition, estimating the prediction intervals under normality assumptions and training many neural networks is computationally expensive. Further, different gradient-boosting-based algorithms are proposed to incorporate uncertainty in prediction [15,16].

In addition, Segù et al. [10] proposed Bayesian belief networks and Monte-Carlo-sampling-based uncertainty prediction techniques that consider prediction uncertainty and sensor noise. Blundell et al. [17] introduced a new method called Bayes by Backprop for model uncertainty estimation in NNs. Unlike most other NNs, Bayes by Backprop estimates the probability distribution of weights (or parameters) instead of estimating fixed values. The interval of an output variable is estimated from the probability distribution of weights. Bayesian models offer a mathematically grounded framework to reason about model uncertainty, but with a high computational cost [18]. A model uncertainty estimation method known as Monte Carlo (MC) dropout is proposed by Gal and Ghahramani [18]. The authors showed that the dropout approach could be interpreted as a variational approximation to the posterior of a Bayesian neural network. It is also a Bayesian approximation of Gaussian processes. Srivastava et al. [19] proposed the standard dropout technique for reducing overfitting. MC dropout is used to predict the probability distribution of a target variable from the standard dropout technique without increasing computational expense.

Although a few methods have been developed to account for uncertainty in data analysis, uncertainty is rarely included in manufacturing decision-making. Uncertainty in the real-world manufacturing environment is unavoidable [20,21]. Therefore, it is inevitable to incorporate uncertainty to make a robust decision and risk assessment in a manufacturing environment. A manufacturing environment has several factors for uncertainty, such as variable loads and working conditions. These uncertainty factors may affect sensor data for condition monitoring. Tool wear prediction under uncertainty is essential to improve maintenance decisions by providing in-depth insight into machine conditions. Different machine learning algorithms are applied to predict tool wear from sensor data, but the sources of uncertainty are rarely considered for tool wear prediction. The deterministic prediction does not always represent actual real-life situations. The reason is that the tool wear process is considered a stochastic process [22].

It is rarely applied to data-driven methods that incorporate uncertainty in manufacturing-related decision-making. Wang et al. [3] demonstrated a probabilistic method based on particle filtering to incorporate uncertainties for tool wear estimation. Karandikar et al. [22] proposed a Bayesian inference approach to estimate tool life using a random walk method. Ren et al. [23] employed a Type-2 fuzzy approach to account for uncertainty in tool wear prediction. An interval-based technique was proposed by Pal et al. [24] to evaluate the robustness of NNs to uncertain input data. The proposed method was applied to a turning operation data set. The authors showed that a back-propagation neural network is more robust than a radial basis function network to the data uncertainty. A MapReduce-based parallel random forest algorithm was implemented to estimate the tool wear as intervals of a milling cutter and reduce the computational time of the original random forest algorithm [25].

A good standard or recommended method for considering uncertainty in tool wear prediction has not been established yet. Research on tool wear prediction under uncertainty is still in the beginning stage. Thus, a novel method for accounting uncertainty during tool wear prediction is proposed in this paper as a contribution to the research area under the umbrella of data-driven tool wear prediction. The proposed method is able to predict several values of a target variable (e.g., tool flank wear) for a new observation (e.g., sensor data) by applying a random dropout approach to a trained NN model. From the several predicted values, the prediction interval can be estimated.

The proposed approach considers data and model uncertainty to make predictions that employ the updated NN with the MC-dropout approach. In the proposed MC-dropout-based prediction method, an empirical mode decomposition (EMD) is used to denoise the collected sensor signals to reduce data uncertainty. EMD improves data quality by reducing noise arising from various sources, such as sensor errors, signal conditioner noise, or environmental noise. Further, time-domain statistical features are extracted from the denoised signals. The principal component analysis (PCA) is used to reduce features and generate linearly independent features. Prediction accuracy and model stability increase while computation cost decreases when the number of features is low, and the features are independent [26]. Finally, MC dropout is applied to consider model uncertainty and predict tool wear as an interval instead of a single-point estimate. An NN model can predict different estimates for a given input observation by applying MC dropout. The probability distribution and interval of the tool flank wear can be estimated from these different values. The range of an interval represents the degree of uncertainty in the prediction of tool wear. Given the manufacturing environment has several sources of uncertainty, interval estimate of tool wear prediction instead of point estimation is preferable to understand the condition of machines and facilitate more realistic maintenance decision-making. Comparative analysis with other methods using various performance metrics shows that the proposed approach achieves higher prediction accuracy.

The paper primarily focuses on tool wear analysis, utilizing data from a milling machine to develop and train the proposed method. However, it is essential to note that the versatility of this approach extends beyond tool wear analysis. The method presented here is adaptable to various condition-monitoring scenarios and can effectively address other types of malfunctions. To apply this approach to different situations, it is necessary to employ suitable sensors that align with the specific data requirements and user objectives. These sensors are instrumental in collecting the necessary data, which are then used to train a data-driven algorithm designed to predict a target variable. Once the model is trained, it can be deployed to continuously monitor and assess the conditions of equipment or systems in real-time, making it a valuable tool for proactive maintenance and fault detection across diverse industrial applications.

The main contribution of this paper lies in its novel data-driven method, effectively managing uncertainty for improved robust tool wear prediction. The highlights of major contributions are as follows:

1. The data and model uncertainty are incorporated for tool flank wear estimation.
2. Different values of tool flank wear are estimated for a new observation by applying MC dropout.
3. Tool flank wear is estimated as intervals from the mean and standard deviation of the predicted values to incorporate uncertainty.

The rest of this paper is organized as follows: The steps of the proposed tool wear prediction method are described in Section 2. In Section 3, the proposed method is implemented in a milling machine data set to demonstrate its applicability and performance. Finally, discussions and conclusions are presented in Section 4.

## 2. Proposed Dropout-Based Prediction Method

Tool wear can be defined as the gradual degradation of cutting tools caused by regular operation [27]. Continued tool wear causes an increase in cutting forces, cutting zone temperature, and the chance of tool breakage, which leads to poor surface quality, low dimensional accuracy, and damage to machine tool components and workpieces. Among different types of tool wear, tool flank wear is a vital tool life criterion and significantly impacts workpiece quality [28,29]. This paper proposes a data-driven framework incorporating uncertainty for interval tool flank wear estimation, as shown in Figure 1.

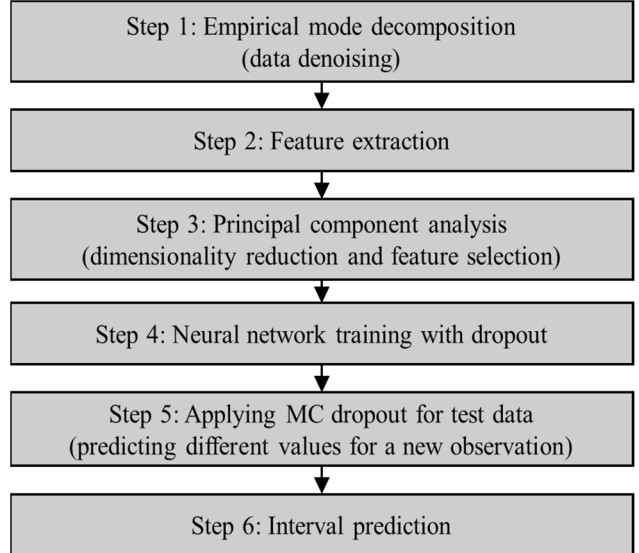

**Figure 1.** A step-by-step process for the proposed framework.

The sensor data collected during the machining process are used to develop a tool wear (tool flank wear) prediction model. In the first step, the collected sensor data are denoised using EMD to reduce data uncertainty. In the next step, a total of eleven (11) time-domain statistical features are extracted from the denoised sensor data. After feature extraction, PCA is used to reduce the dimensionality and convert data with linearly independent features. Further, an NN model with a dropout approach is trained using the new data obtained from PCA. In the next step, the dropout approach is applied to a new observation known as the MC dropout to obtain multiple tool wear values. In the last step, the interval of tool flank wear can be determined for the new observation from the predicted values. The details of each step are demonstrated in the following subsections.

*2.1. Empirical Mode Decomposition*

The uncertainty present in data can be reduced by applying different data-denoising algorithms. The EMD method introduced by Huang et al. [30] is a well-known algorithm used for data denoising. It decomposes a signal into a collection of intrinsic mode functions (IMFs) and a final residual [31]. An IMF must satisfy both of the following two criteria:

1.  For a given signal vector, the number of extrema and the number of zero crossings must either be equal or differ by at most one.
2.  At any point, the mean value of the envelope defined by the local maxima and the local minima is zero.

A subset of the IMFs from all decomposed IMFs still consists of noise, and therefore, the goal is to identify and subtract those IMFs from the original signals to obtain noiseless signals. The signal decomposition algorithm to extract IMFs is given in Figure 2.

The extraction of IMFs is stopped when one of the following conditions is satisfied: (1) a predefined number of IMFs are being extracted, or (2) the residual becomes monotonic from which no more IMF can be extracted [32]. In this paper, it is assumed that the predefined maximum number of IMFs extracted is n = 10.

When all IMFs are extracted, the signal can be expressed as,

$$x(t) = \sum_{i=1}^{m} IMF_i(t) + r_m(t) \tag{1}$$

where $m$ is the number of IMFs extracted from the original signal $x(t)$; $r_m(t)$ is the residual obtained after extracting the $m^{th}$ IMF.

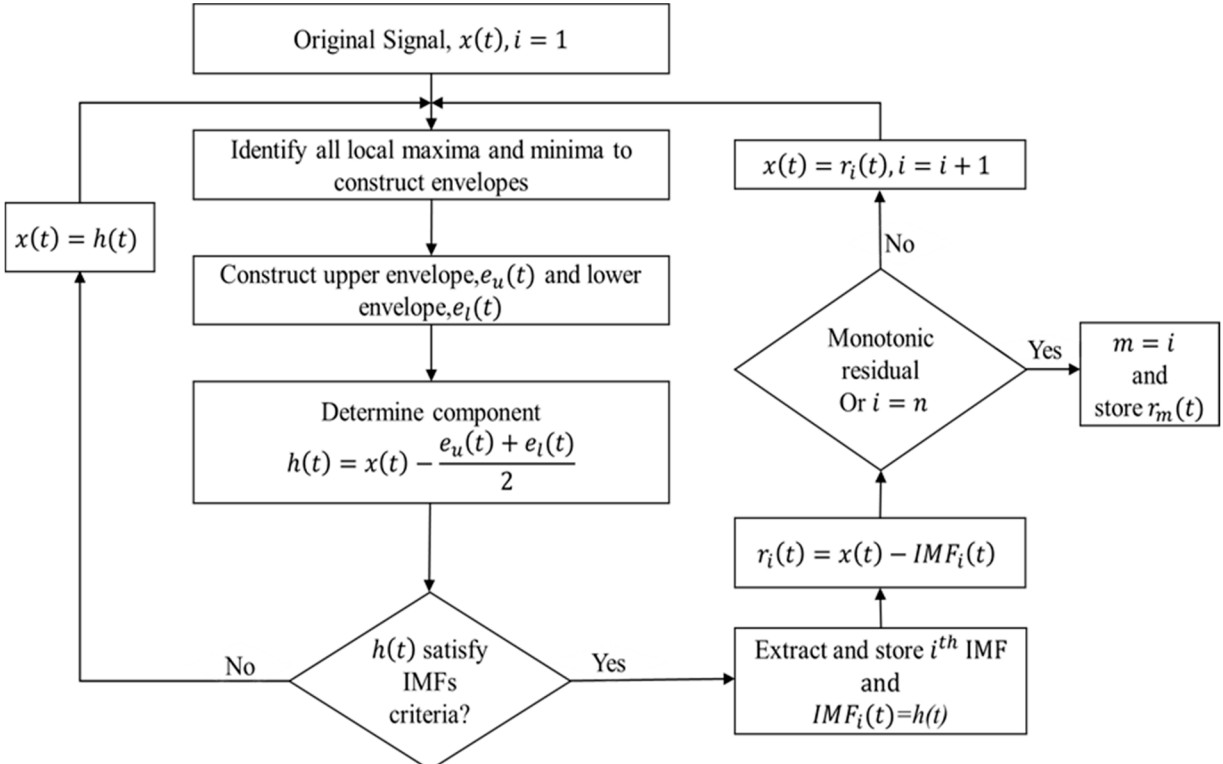

**Figure 2.** A flow diagram of EMD.

Consider the original signal $x(t)$ is a collection of the noiseless signal $\widetilde{x}(t)$ and noise $\eta(t)$ as

$$x(t) = \widetilde{x}(t) + \eta(t) \tag{2}$$

The target is to estimate the denoised signal $\widetilde{x}(t)$ from the original signal $x(t)$ by removing the noise $\eta(t)$. The first IMF contains high-frequency terms in the decomposed signal, and the last IMF contains low-frequency terms [33]. It is also well-established and well-proven that high-frequency terms consist of more noise compared to low-frequency terms.

Consider the first $k$ IMFs consist of noise. Therefore, $\widetilde{x}(t)$ can be written as,

$$\widetilde{x}(t) = x(t) - \sum_{i=1}^{k} IMF_i(t) \tag{3}$$

The value of $k$ can be determined by the correlation coefficient, $\sigma$ which is defined as,

$$\sigma = \frac{x(t)^T \widetilde{x}(t)}{\sqrt{x(t)^T x(t)} \sqrt{\widetilde{x}(t)^T \widetilde{x}(t)}} \tag{4}$$

Assume the threshold value of $\sigma$ is $\rho$. Then, the value of $k$ can be determined by,

$$k^* = \max\{k \mid \sigma \geq \rho\} \tag{5}$$

Once $k^*$ has been determined, $\widetilde{x}(t)$ is further estimated from Equation (3) by considering $k = k^*$. Generally, the threshold value of $\sigma$ is assumed to be between 0.75 and 0.85 [33]. In this paper, the EMD technique is applied to improve prediction accuracy by reducing data uncertainty. The used value of $\rho$ is 0.8.

### 2.2. Feature Extraction

Feature extraction is a process by which an initial raw data set is converted into more manageable groups of features for processing. The time-domain, frequency-domain, and time-frequency domain features can be extracted from a data set. Further data processing is required to extract the frequency-domain features [34,35]. A high computation effort is necessary to extract time-frequency domain features from denoised data [36]. In this paper, 11 time-domain features are extracted from denoised data. The mathematical models for all time-domain features are given in Table 1.

**Table 1.** Time-domain features extracted from denoised data.

| Features | Formula | Features | Formula |
|---|---|---|---|
| 1. Mean | $\mu = \frac{1}{N_s} \sum\limits_{i=1}^{a} \tilde{x}_i$ | 7. Crest factor | $f_c = \frac{\max\left(\tilde{x}_i\right)}{rms}$ |
| 2. Standard deviation | $\sigma = \sqrt{\frac{\sum_{i=1}^{N_s}\left(\tilde{x}_i - \mu\right)^2}{N_s - 1}}$ | 8. Shape factor | $f_s = \frac{rms}{\frac{1}{N_s}\sum_{i=1}^{N}\left|\tilde{x}_i\right|}$ |
| 3. Root mean square | $rms = \sqrt{\frac{1}{N_s} \sum\limits_{i=1}^{N_s} \tilde{x}_i^2}$ | 9. Impulse factor | $f_i = \frac{\max\left(\tilde{x}_i\right)}{\frac{1}{N_s}\sum_{i=1}^{N}\left|\tilde{x}_i\right|}$ |
| 4. Square mean root | $smr = \left(\frac{1}{N_s} \sum\limits_{i=1}^{N_s} \sqrt{\left|\tilde{x}_i\right|}\right)^2$ | 10. Marginal factor | $f_m = \frac{\max\left(\tilde{x}_i\right)}{smr}$ |
| 5. Skewness | $f_{sk} = \frac{\sum_{i=1}^{N_s}\left(\tilde{x}_i - \mu\right)^3}{(N_s - 1)\sigma^3}$ | 11. Peak to peak | $f_{pp} = \max\left(\tilde{x}_i\right) - \min\left(\tilde{x}_i\right)$ |
| 6. Kurtosis | $f_{sk} = \frac{\sum_{i=1}^{N_s}\left(\tilde{x}_i - \mu\right)^4}{(N_s - 1)\sigma^4}$ | | |

### 2.3. Principal Component Analysis

The irrelevant and redundant features increase computational costs and reduce model stability. Principal component analysis (PCA) is one of the most widely used dimensionality reduction techniques that captures the presence of significant variability in the data set and minimizes the loss of information. The following steps provide a detailed discussion of the PCA method:

Step 1: Standardized data by subtracting the mean of features from all observations of the features.

$$\tilde{X} = X - 1^n \overline{X} \tag{6}$$

where, $X \in R^{n \times p}$ is the feature matrix (data) obtained by feature extraction, $\overline{X} \in R^{1 \times p}$ is the mean vector of features, and $\tilde{X} \in R^{n \times p}$ is the standardized data. $N$ and $p$ are the number of observations and the number of features, respectively.

Step 2: Compute the covariance matrix, $S \in R^{p \times p}$ of X.

$$S = \tilde{X}^T \tilde{X} \tag{7}$$

where, $\tilde{X}^T$ represents the transpose of the matrix $\tilde{X}$

Step 3: Compute the eigenvalues and eigenvectors of $S$. In PCA, eigenvectors are the principal components. Then, order the eigenvectors according to the descending order of eigenvalues.

$$V = \begin{bmatrix} v_1 & v_2 & v_3 & \cdots & v_p \end{bmatrix} \tag{8}$$

where, each column $V \in R^{p \times p}$ represents an eigenvector. $v_i(1, 2, \cdots, p)$ is the eigenvector corresponding to the ith largest eigenvalue, $\lambda_i(\lambda_1 \geq \lambda_2 \geq \cdots \geq \lambda_p)$. In eigenspace, $v_1$ represents the maximum direction along the data variance.

Step 4: Set a threshold value $\tau$ of the explained variance and determine a new dimension, $d^*$, using the following formula.

$$d^* = \left\{ \min d \left| \frac{\sum_{i=1}^{d} \lambda_i}{\sum_{i=1}^{p} \lambda_i} \geq \tau \right. \right\} \tag{9}$$

where, $d^*(d^* \ll p)$ is the reduced dimension.

Step 5: Determine the new matrix, $V^* \in R^{p \times d^*}$, by taking the first $d^*$ columns of $V$.

$$V^* = \begin{bmatrix} v_1 & v_2 & v_3 & \cdots & v_{d^*} \end{bmatrix} \tag{10}$$

Step 6: Finally, the reduced data set, $X^* \in R^{n \times d^*}$, is computed by the formula given below.

$$X^* = \tilde{X}V^* \tag{11}$$

The data set $X^*$ obtained by PCA is then used to train the NN employed in the tool flank wear prediction. In this paper, PCA is applied to the extracted features matrix to reduce the data dimensionality.

### 2.4. Overview of Neural Network

The NN approach is widely used as a prediction approach in different areas, including manufacturing, medicine, supply chain, and many others. In an NN, a neuron of a layer is fully connected with all neurons of prior and post layers, as shown in Figure 3a.

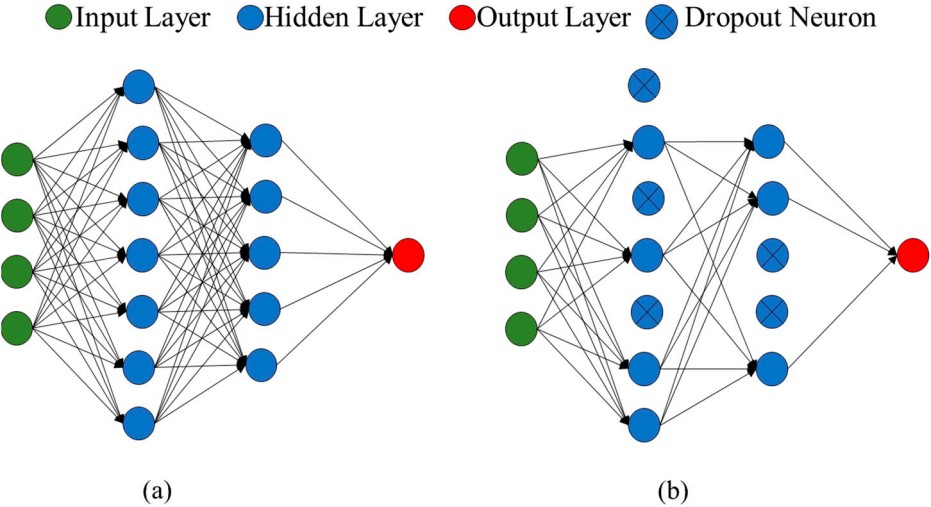

(a)                                    (b)

**Figure 3.** (**a**) Standard NN and (**b**) NN after applying dropout.

There is no standard rule for selecting the architecture of neural networks. For all data-driven models, it is essential to consider bias and variance for model selection. A well-fitted model can predict with high accuracy when bias and variance are low. Both bias and variance can be reduced by increasing training data size, but it is not always practical due to data scarcity. In addition, the bias can be reduced by selecting a complex architecture of neural networks, and the variance can be minimized by using different regularization techniques. In this research, numerous experiments were conducted to determine the appropriate model, model parameters, and model architecture. Initially, the experiments began with a modest neural network featuring only one hidden layer to establish a baseline for model architecture selection. Subsequently, the complexity of the architecture was systematically augmented by introducing additional hidden layers and increasing the neuron count within each layer. This iterative process was continued until diminishing prediction accuracy and the onset of model overfitting were observed,

thereby allowing for the optimal configuration to be identified. The final architecture will be detailed in Sections 3.2 and 3.3.

A great statistician of the 20th century, George Box, mentioned that all models are wrong, but some are very useful [37]. An NN approach is considered a particularly useful model known to predict with high accuracy. In most cases, a prediction obtained by an NN is a deterministic point value, and it is assumed that the level of prediction accuracy is relatively high. This deterministic prediction is extremely optimistic, and the model uncertainty is often ignored. Thus, it is essential to know the level of uncertainty in a prediction to make robust decisions with high confidence. This paper employs the MC dropout approach to account for model uncertainty in predictions.

### 2.5. Monte Carlo (MC) Dropout

Srivastava et al. [19] proposed a standard dropout technique to prevent overfitting and efficiently combine different NN architectures. Dropout can be defined as deleting a neuron from networks and all its associated connections (Figure 3b). The dropout decision is binary; it is whether a neuron is retained in an NN or is dropped out from an NN. If the dropout probability is $p$, then the probability of a neuron retaining in the network is $(1 - p)$. In the traditional dropout, a fully connected network is applied to predict the target variable for a new observation. In addition, the final weight is estimated as $(1 - p)w$ for applying a fully connected network. However, in MC dropout, the dropout is applied to the trained network for test data to know the uncertainty and confidence. The MC process is as follows.

1. The dropout is applied at probability, $p$.
2. When training an NN, the weights of the retained neurons update in each epoch, and the weights of dropped-out neurons remain unchanged.
3. Dropout is applied randomly until the desired level of accuracy is obtained.
4. After training, the dropout is applied to the trained NN to predict different values for a new observation. This concept is known as MC dropout.

For a new observation, the dropout is applied to a trained NN to predict multiple values to obtain a distribution of a target variable. When dropout is applied in an NN several times, some neurons are masked out randomly. For a new observation, several values are predicted using different network architectures. From the values of the target variable, the distribution is estimated. More details, including theoretical proof of MC dropout, can be found in Ref. [18]. In this research, MC dropout is applied for a new observation to estimate tool flank wear as an interval.

### 2.6. Interval Prediction

When an NN model predicts several values of tool flank wear for a new observation, the first two moments (mean and variance) of the tool flank wear can be determined from the predicted values ($\hat{y}_i$) by following models [11].

$$\hat{\mu} = \frac{1}{D}\sum_{i=1}^{D}\hat{y}_i \tag{12}$$

$$\hat{\sigma}^2 = \frac{1}{D-1}\sum_{i=1}^{D}(\hat{y}_i - \mu)^2 \tag{13}$$

where dropout is randomly applied $D$ times for a new observation. When $\mu$ and $\sigma$ are known, the prediction interval (PI) for a new observation for a certain level of significance ($\alpha$) can be estimated as follows.

$$PI = \begin{bmatrix} \mu - Z_{\alpha/2}\sigma & \mu + Z_{\alpha/2}\sigma \end{bmatrix} \tag{14}$$

where $PI$ is the prediction interval and $Z_{\alpha/2}$ is the $z$-value for $\alpha$ significance level. The range of PI represents the level of uncertainty. A narrower range of PI indicates a lower level of uncertainty, and a wider range represents a higher level of uncertainty. Based on the level of uncertainty, the maintenance decision will be taken to reduce unwanted failure. Therefore, interval prediction by considering data and model uncertainty provides more detailed information as a basis for making decisions like maintenance scheduling and tool changing. In the following section, the proposed tool flank wear prediction method is implemented on a classic numerical data set to assess the performance of the proposed dropout-based method.

## 3. Numerical Experiments and Results

In this section, the proposed method is applied to the 2010 Prognostic and Health Management (PHM) Challenge Data Set [38] to demonstrate the performance and applicability of the proposed method. The NN model is first trained, and then the interval tool flank wear is predicted for the test data according to the proposed framework.

### 3.1. Data Description

In this paper, a collected data set is used to demonstrate the proposed method. The data collection procedure is detailed in this section. Seven sensors were installed on the Röders Tech RFM760 CNC milling machine to collect tool condition data of a tungsten carbide cutter with three flutes [38]. Among the seven sensors, three force sensors are used to collect signals for force along the X, Y, and Z axes, and three vibration sensors used to collect signals for vibrations along the X, Y, and Z axes. The seventh sensor was an acoustic emission (AE) sensor, and other machining parameters assumed constant. Tool flank wear was set as the target variable and measured for sensor data. The data from six cutters ($c_1, c_2, c_3, c_4, c_5$, and $c_6$) were collected. A total of 315 observations were collected for each cutter. The tool flank wear was measured for three cutters, ($c_1, c_4$, and $c_6$) and the measurement unit is μm. Tool wear for all three flutes was measured for the three cutters, and the average tool flank wear was used as the target variable in this paper. The measured tool flank wear for the three cutters is shown in Figure 4.

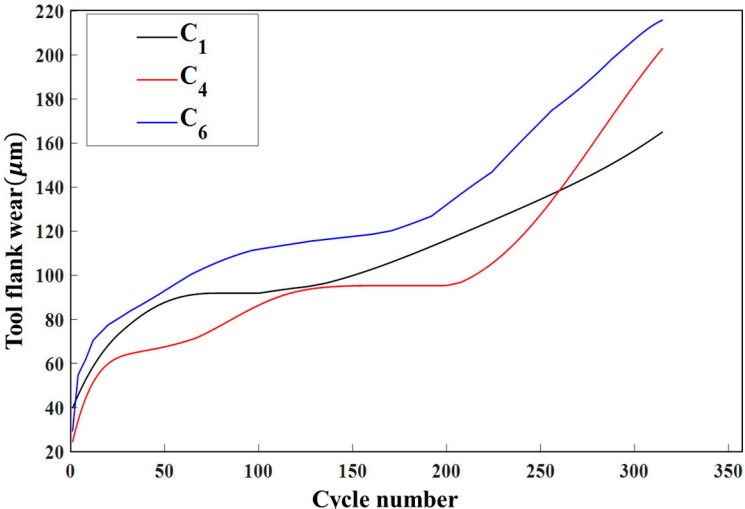

**Figure 4.** Measured tool flank wear.

Although the machining parameters are the same for all three cutters, the tool flank wear patterns are different for each cutter. Therefore, deterministic assumptions may not be useful for this case because capturing the exact pattern of the tool flank wear from a specific cutter is challenging and may not be applied to another cutter. The sensor data of the cutters with known tool flank wear are used in this paper. Among three cutters with known tool flank wear, the data of cutters $c_1$ and $c_4$ are used to train the NN with MC

dropout, and the data of the third cutter $c_6$ is used as test data to predict interval tool flank wear. The data descriptions are summarized in Table 2.

**Table 2.** Data Descriptions.

| | |
|---|---|
| Sensor type | X-axis force<br>Y-axis force<br>Z-axis force<br>X-axis vibration<br>Y-axis vibration<br>Z-axis vibration |
| Total extracted features | $7 \times 11 = 77$ |
| Target variable | Tool flank wear |
| Data size | Cutter $C_1$ : 315 datapoints<br>Cutter $C_4$ : 315 datapoints<br>Cutter $C_6$ : 315 datapoints |
| Training data size | Cutters $C_1$ and $C_4$: 630 data points |
| Test data size | Cutter $C_6$: 315 datapoints |

*3.2. Tool Flank Wear Prediction*

In the first step, the data collected from seven (07) sensors are denoised by EMD. The IMFs extraction process is ended if the maximum number ($n = 10$) of IMFs has been extracted or the residual becomes monotonic for a smaller number of IMFs. The threshold value of the correlation coefficient is $\rho = 0.8$ for removing IMFs that consist of noise. After data denoising, the 11 time-domain features listed in Table 1 are extracted from the denoised data of each sensor. As there are seven (07) sensors, a total of 77 ($11 \times 7$) features are extracted from all denoised sensor data for each observation. Thereafter, the PCA is used to reduce data dimensionality and generate independent features. For this data set, the dimension is reduced to $d^*$ number of features that can explain at least 95% ($\tau = 0.95$) variation. The extracted 77 features are reduced to $d^* = 25$ features that capture 96.7% variation. These new 25 features are then applied to train the NN.

The reduced data set with 25 features obtained after applying PCA is used to train the NN with MC dropout for interval prediction. This paper considers an NN with two hidden layers to capture non-linearity. In this research, the architecture of neural networks is chosen by trial and error. Different combinations of hidden layers, the number of neurons in hidden layers, and activation functions are applied to select a model architecture that minimizes the loss function and increases prediction accuracy. The selected model architecture is discussed as follows: the activation function used is the rectified linear unit (ReLU) for the hidden layers, and the number of neurons in the hidden layers is 20 and 16. A dropout rate of 0.2 ($p = 0.2$) is chosen for NN models. Adam optimizer [39] is used to train the NN with a learning rate of 0.001, exponential decay control parameters of 0.99 and 0.999, and a smoothing term of $10^{-8}$. Unlike the standard dropout technique, multiplying weights with probability to obtain the final weights, MC dropout is applied several times to new observations of the cutter $c_6$ data for predicting different values of tool flank wear. The confidence interval is determined from all predicted values of tool flank wear.

When new sensor data are available, the data are initially denoised by EMD to reduce data uncertainty. Then, all 77 features are extracted from the data of all seven sensors. Then, by using $V^*$, the dimensionality of the data is reduced to $d^* = 25$. The matrix $V^*$ is obtained from training data when PCA was applied for feature reduction. Different tool flank wear prediction values can be obtained by applying the dropout technique 50 times (D = 50) on the trained NN for new data. The mean and standard deviation of all estimated values of tool flank wear are determined using Equations (12) and (13), respectively. Finally, the 95% confidence interval is estimated using Equation (14) around the predicted tool wear

value, capturing uncertainty in estimated values. The interval prediction using a neural network and MC dropout is demonstrated in Figure 5.

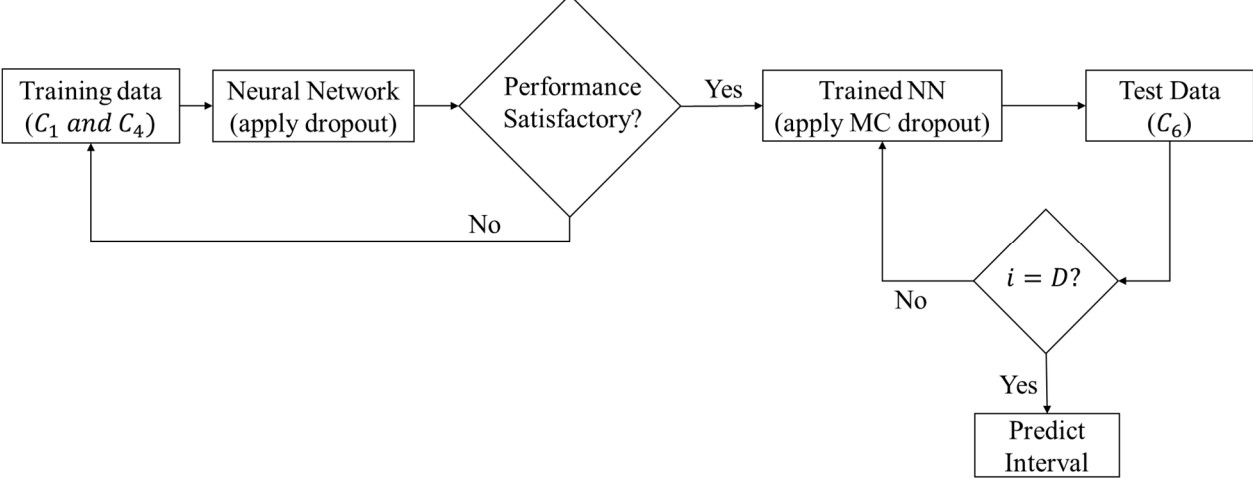

**Figure 5.** Interval tool wear prediction.

*3.3. Results Analysis*

The result of the estimated tool flank wear interval with the MC dropout is shown in Figure 6. Tool wear of the cutting tool can be characterized by three failure modes, namely, a break-in region (rapid wear), a steady-state region (uniform wear), and a failure region (rapid wear) [40]. The prediction accuracy of tool flank wear is different for different regions. The mean values of predicted tool flank wear are poor or less accurate for the break-in region. The majority of research on tool wear prediction by data-driven methods has shown that prediction accuracy is high for the failure region [41].

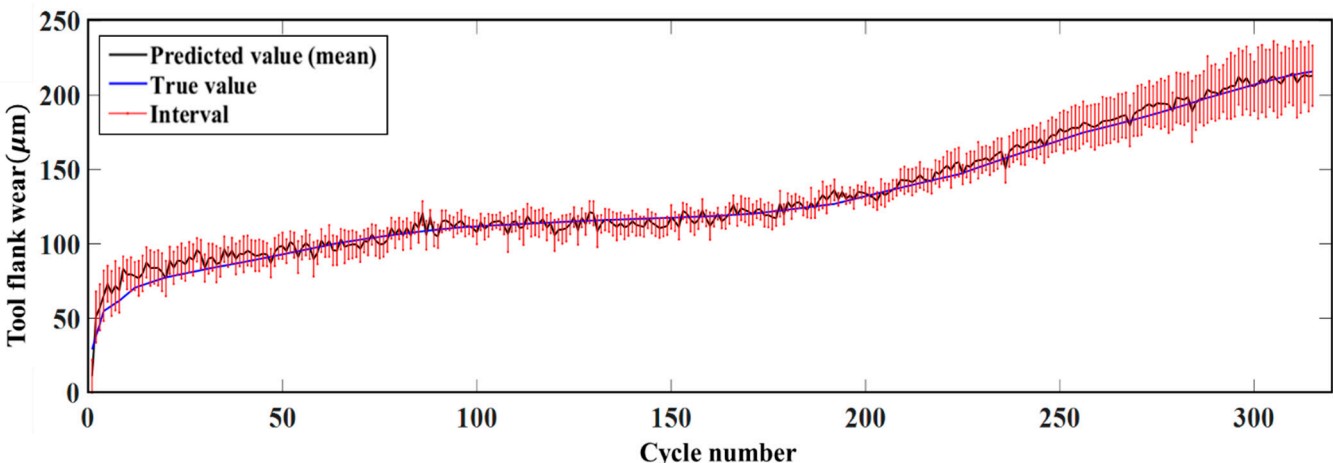

**Figure 6.** Interval prediction result for tool flank wear.

It is visible that the range of the intervals is different for different points in the three regions, as expected. The predicted interval of the tool flank wear is small for the steady-state region as the chance of tool breakdown is low in the region. It indicates that uncertainty is relatively low. The chance of tool failure is high for the later cycles, approximately after the 250th cycle—the failure region, demonstrated in Figure 6. Accurate predictions of tool flank wear are essential for these later cycles to estimate an optimum tool changing time and make other maintenance decisions to avoid significant downtime. Tool wear accelerates rapidly in the failure regions. It is observed that the prediction intervals are comparatively wider for the last few cycles (after the 290th cycle in Figure 6). Tool wear

accelerates rapidly in the failure regions, and sudden failure may occur in the region. For this, the prediction interval is more comprehensive for the failure region.

Another reason for this is the range of the tool wear is different for test and training data. The maximum tool flank wear in the training data is 203.08 mm, whereas the maximum tool flank wear in the test data is 215.94 mm. For some observations in the test data, sensor signals are different, and measured tool flank wear is higher than the maximum tool flank wear in training data. The resulting interval is comparatively wider for these test observations because the uncertainty is higher for extrapolation [18].

Standard regression and classification algorithms did not capture uncertainty for point value prediction. The prediction ignoring uncertainty is applied to support different decision-making. In such cases, it is assumed that a model is highly reliable and can predict with high confidence, but this is not always true [42]. An input may be subject to noise and outliers, and model uncertainty may arise due to model parameters and architecture assumptions [43,44]. Interval prediction is a way to represent these uncertainties in quantitative form. For example, a 95% prediction interval for a new observation can be defined as 95% confidence that the true value for the observation will fall within the upper and lower bounds of the interval. The mean value is the most likely predicted value, and the prediction interval determines the variability of prediction around the mean and the level of uncertainty. A wider prediction interval represents the high prediction variability around the mean predicted value. A wider prediction interval is a less reliable prediction, but it is useful for analyzing a system as it explains the condition of a system instead of solely relying on optimistic point value estimation as the point value estimation fails to explain variability. A narrower interval is preferable and indicates a low level of uncertainty.

Additionally, it is also vital to use high-quality data and select appropriate data preprocessing techniques along with machine learning algorithms. The prediction accuracy may decrease due to noisy data, high data dimension, and inappropriate preprocessing algorithms. In this research, Pearson correlation is used as a feature reduction technique instead of PCA to analyze the impacts of PCA. It is named Model 1. Another model (Model 2) is trained using raw data instead of denoised data to investigate the impact of EMD on prediction accuracy. The details of Models 1 and 2 will be explained in the following sections. The performance of the models is quantified through the mean absolute error (MAE), root mean square error (RMSE), and R-squared ($R^2$) values. The formulae for determining performance metrics are demonstrated in Table 3. In these formulas, $y_i$ is the actual tool flank wear, $\hat{y}_i$ is predicted tool flank wear, and $\overline{y}$ is the average of actual tool flank wear.

**Table 3.** Performance Matrices.

| Performance Matrices | Formulae |
|---|---|
| MAE | $MAE = \frac{1}{n}\sum_{i=1}^{n}|y_i - \hat{y}_i|$ |
| RMSE | $MSE = \frac{1}{n}\sum_{i=1}^{n}(y_i - \hat{y}_i)^2$ |
| $R^2$ value | $R^2 = 1 - \frac{\sum_{i=1}^{n}(y_i - \hat{y}_i)^2}{\sum_{i=1}^{n}(y_i - \overline{y})^2}$ |

For this, in a model (Model 1), the Pearson correlation matrix is applied for feature selection instead of PCA. The correlation of all features with the target variable is determined, and a feature is selected to train the neural network model if the linear correlation is more than 0.75. A total of 37 features are selected after applying the Pearson correlation. Model 1 has one hidden layer with 16 neurons. The used activation function is ELU, and the dropout rate is 0.2 for the hidden layer. All other parameters are retained unchanged. The denoised data are applied for Model 1. Comparison between Model 1 and the proposed model is performed to analyze the impacts of feature reduction techniques on the

performance of three metrics. Table 4 provides the summary of the comparison of all three models considering RMSE, MAE, and R$^2$ values. The comparative analysis shows that the proposed feature selection method provides significantly better performance on all three metrics. The proposed method's RMSE, MAE, and $R^2$ values are better than that of Model 1, which used the Pearson correlation for feature selection. For PCA, when compared to the Pearson correlation, the performance is improved by up to 35%. It can be summarized that appropriate feature reduction technique selection is vital for prediction accuracy. For the data set, the performance of using PCA is better than the Pearson correlation.

**Table 4.** Comparison of different methods.

|  | RMSE | MAE | R$^2$ Value |
|---|---|---|---|
| Model 1 | 0.183 | 0.307 | 0.82 |
| Model 2 | 0.083 | 0.238 | 0.913 |
| Proposed method | 0.062 | 0.175 | 0.937 |
| Ref. [45] | 0.908 | 0.440 | - |
| Ref. [46] | 0.0212 | 0.0146 | 0.89 |

In Model 2, the raw data without denoising are used to investigate the impacts of EAD, the data denoising technique, on the prediction accuracy of neural networks. In addition, the PCA is applied to select linearly independent features like the proposed method. Two hidden layers with 16 and 10 neurons are used to build neural architecture for Model 2. ELU activation function and 0.2 dropout rate are used for both hidden layers. All other variables are the same as the proposed model. For Models 1 and 2, the model architecture is chosen by trial and error, and the best architectures with high prediction accuracy are compared with the proposed model. Model 2 is trained, and three performance metrics are evaluated. RMSE, MAE, and R$^2$ values are given in Table 4 for Model 2. All three metrics of Model 2 are better than that of Model 1. It can be concluded that the feature selection technique is significant compared to the data denoising technique for the data set.

In addition, it is clear from Table 4 that all performance metrics are better for the proposed model than Model 2. Therefore, it can be concluded that data denoising improves prediction accuracy. Overall, the performance of the proposed method is still superior to the two other models. The combination of preprocessing techniques, EMD, PCA, and the MC dropout employed in the proposed framework contributes to the prediction performance. Therefore, the data denoising techniques, feature section methods, and mode architecture are vital for prediction accuracy along a data-driven algorithm. It can be further concluded that the proposed method is more suitable for practical cases as the proposed method predicts intervals instead of point values only.

In addition, the proposed method is compared with Refs. [45,46]. Both articles trained LSTM networks but used different model architectures, model parameters, and performance metrics. Ref. [45] only estimated MAE and RMSE. For this, both matrices are used to compare with the proposed model. From Table 4, the proposed method outperforms in terms of MAE and RMSE. In another comparison, it is shown in Table 4 that the $R^2$ value of the proposed methods is better than Ref. [46]. Although the proposed method is compared with two published articles, the current research is unique as the goal is to estimate intervals by incorporating data and model uncertainty. No comparison is performed for interval prediction because, to the best knowledge of authors, no research exists that has proposed to predict interval estimates.

## 4. Discussions and Conclusions

In the manufacturing environment, uncertainty is a common phenomenon, and tool wear prediction incorporating uncertainty is beneficial for making crucial decisions that significantly impact financial performance. This paper proposes the MC-dropout-based prediction framework for tool flank wear prediction under uncertainty. One of the unique features of the proposed method is that it considers both model and data uncertainties

to predict intervals. The data noise is reduced in the first step of the proposed approach by using the EMD approach. In the second step, for a new observation (sensor data), the tool flank wear is predicted as an interval using MC dropout. PCA is applied for feature reduction and the generation of independent features that reduce computational costs and improve prediction accuracy. Instead of point estimation, the estimated mean value and the degree of uncertainty (interval range) are able to provide more information for making maintenance decisions and determining the tool changing time.

The proposed method is applied to a real-world manufacturing data set to investigate the performance and the applicability of the proposed method. Based on the investigation, the proposed method is useful for predicting an interval that includes the most likely value and the level of uncertainty for a new observation. The interval prediction is unique and significant compared with point value estimation as it incorporates model and data uncertainty. The proposed method is applicable to monitor machine malfunctions in designing maintenance strategies to reduce machine downtime.

In addition to addressing model and data uncertainty, a manufacturing environment introduces various other sources of uncertainty, including environmental fluctuations, variable workloads, inherent randomness, and potential sensor errors. Enhancing prediction accuracy and robustness necessitates comprehensively considering these diverse uncertainty factors. In the future, it is imperative to incorporate other sources of uncertainty with data and model uncertainty into tool wear prediction models. Beyond this, the selection of model architectures and parameters has often relied on a trial-and-error approach, potentially overlooking architectures capable of delivering highly accurate results. To rectify this, a systematic methodology for the model architecture selection process should be developed or adopted. Furthermore, our future research endeavors will explore the integration of frequency-domain and time-frequency domain features. This exploration aims to augment prediction accuracy and ensure a more comprehensive understanding of tool wear dynamics. Finally, the proposed model prediction can be used as automatic input for maintenance management systems, for instance, a simulation model given in Ref. [47]. In the future, software will be developed for automatic maintenance scheduling and feedback for Industry 4.0.

**Author Contributions:** Conceptualization, N.Y. and A.D.; methodology, A.D. and N.Y.; software, A.D.; validation, N.Y., O.P.Y., M.R. and R.S.; formal analysis, A.D, N.Y. and O.P.Y.; investigation, N.Y. and O.P.Y.; resources, N.Y. and O.P.Y.; data curation, A.D. and N.Y.; writing—original draft preparation, A.D., N.Y. and O.P.Y.; writing—review and editing, A.D., N.Y., O.P.Y., R.S. and M.R.; visualization, A.D., N.Y. and O.P.Y.; supervision, N.Y. and O.P.Y.; project administration, N.Y.; funding acquisition, A.D. and O.P.Y. All authors have read and agreed to the published version of the manuscript.

**Funding:** This research is partially funded by the National Science Foundation (NSF) EPSCoR RII Track-2 Program under the NSF award # 2119691. and NASA SPICES project under award# 80NSSC22K1427.

**Data Availability Statement:** The data presented is available upon request.

**Conflicts of Interest:** The authors declare no conflict of interest.

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
