# Peer review of "Addressing Uncertainty in Tool Wear Prediction with Dropout-Based Neural Network"

_computers, doi:10.3390/computers12090187_

Round 1
Reviewer 1 Report
Overall, the paper seems good. However, I have the following concerns:
- The authors compared their proposed NN model against two other NN models. However, the authors need to provide more information on this comparison.
- The authors need to explain the reasons behind the changes made to the other two models.
- Pearson correlation is a feature selection technique, while PCA is a dimensionality reduction technique. Directly comparing them is not reasonable. (Authors may try other dimensionality reduction techniques.)
- The authors need to list the changes they made to the other two models. It is not easy to understand what was changed and what wasn't.
- Authors may compare the proposed model with other models from other studies. Since the comparison used standard metrics like MAE and R2 instead of interval prediction, other models can be used to compare the performance of the proposed model.
The quality of writing/English of this paper needs moderate amount of editing.
Author Response
Dear Reviewer,
We appreciate your great efforts in reviewing the manuscript in detail. We have carefully revised the manuscript and addressed all the concerns and comments being raised in the previous version. The responses to the comments are detailed in the attached file.
Sincerely
Authors

Reviewer 2 Report
Dear Authors!
You have touched on a very interesting and useful topic.
Diagnostics of equipment malfunctions, as well as the development of methods and models using the provisions of artificial intelligence, aimed at improving the accuracy of the forecast, are of both theoretical and practical interest, in particular, among developers of AI engineering applications.
The approach is more than clear. I regard the choice of the mathematical apparatus as successful.
However, despite the overall positive impression, there are a number of points that I would like to draw your close attention to:
1. There are many reasons for malfunctions (tool wear), as you noted, but they also arise due to the principle of equipment operation.
I did not find in your work an assessment of the proposed solutions for different strategies for operating the equipment.
If we are talking only about the regular mode, then the value of the study narrows ...
2. Section 3.2 describes the resulting architecture. Be sure to substantiate it. You obviously did a lot of experiments.
3. As a recommendation, please pay attention to how your colleagues solved a similar problem: Simulation model performance evaluation of repair-diagnostic complex (2016) ARPN Journal of Engineering and Applied Sciences, 11 (16), pp. 9636-9645 - http://www.arpnjournals.org/jeas/research_papers/rp_2016/jeas_0816_4815.pdf
How useful this idea may be for your research, from the point of view of the "integrated module" of the diagnostic system, which increases the accuracy of the prognosis.
4. Section "Results" should contain methodological results.
5. The Discussion section should evaluate the approaches described in the Introduction section.
Good luck with revisions!
Author Response

(The authors gave the same response as above.)

Reviewer 3 Report
-After reading this paper, my conclusion is that the contribution
shown in the manuscript is not significant to be accepted in its current form.
The paper is very well prepared. However, it has little research contribution. The applied procedures are well known and applied by many other researchers in a similar way to a similar problem. The research is not novel.
The paper does not contain new and significant information adequate to justify publication.
-The novelty and contribution is not clear. It is suggested to carefully revise the manuscript to highlight the major contribution in the introduction section and introduce a precise organization of the paper in the last paragraph of the introduction which
may help the reader to follow.
The readers will find a lot of descriptive reports when they finishing their reading. Most of the details are missing in this paper.
The state of the art is not well described, and the knowledge gap is not clearly defined.
The objectives are not well articulated.
-Related studies:
The contrast between different methods, their limitations and the novelty (contribution) of the proposed method is clear.
As the underlying problem in the paper has been studied also by others, so the unique features of the approaches proposed and the main advantages of the techniques over others have to be clearly commented and shown.
Are there any deficiencies of the results and methods in this paper and how to make further improvements, to make your results less conservative?
Most of the material shown in this paper comes from previous studies and it is suggested to remove less important explanations.
The topic of this paper could be important for practice and the outcome could be very useful for journal readers. Keywords seem to be appropriate. The presented paper is reasonably easy to follow and understand. The paper is technically correct. The paper is logically written. I did not find errors. English language and terminology is clear. The paper is readable and will be interesting for journal readers.
The terminology is faultless.
The title is representative of the article's contents.
The abstract summarises the contents clearly.
-introduction; state of the art novelty should be exposed,
- a lot of descriptive report regarding methods. It should be reduced. The main objective of the paper is in the method.
The results are not clearly stated.
The introduction provides a brief overview of the topic but lacks a clear problem statement. It would be beneficial to clearly state the research problem that the proposed model aims to address. The motivation for the research is not well-established. It would be helpful to explain why existing models are insufficient and how the proposed model improves upon them.
The related work section is somewhat limited. Consider expanding it to include a more comprehensive review of existing wear models.
The methodology section lacks sufficient details for readers to understand the proposed model thoroughly. Provide a step-by-step explanation MC.
The evaluation section should include a detailed description of the experimental setup, including the datasets used, evaluation metrics, and baseline models for comparison. Provide more quantitative and qualitative results to demonstrate the effectiveness of the proposed model. Include comparisons with baseline models and statistical significance tests where applicable.
Discuss the limitations of the proposed model and potential avenues for future research. The conclusion should summarize the main contributions of the paper and highlight the key findings.
Visuals: The paper lacks visual aids such as figures, diagrams, or tables, which could enhance the understanding of the proposed model. Consider including visual representations of the model architecture or experimental results, as appropriate.
The work described in this paper is topical and potentially would be of interest to other scholars. But its nature is not suitable for publication in the journal because, compared with the state of the art as exemplified by several recent publications on the general subject area, it does not advance understanding of fundamental neural computing and applications.
To reiterate; the work does not have sufficient scientific depth for publication in this state, but, with suitable revision, it could be acceptable. In summary, this contribution needs major corrections with reasonable care for details.
Author Response

(The authors gave the same response as above.)

Round 2
Reviewer 3 Report
The modified recommendations are accepted, and I found that it also contributed to the applicative strength of the described research. It seems that the majority of recommendations and requests from other reviewers have also been accepted.
The paper is well prepared. However, it has a little research contribution. The studied problem is well known.